Impact of copper toxicity on stone-head cabbage (Brassica oleracea var. capitata) in hydroponics

Ali Sajid 1 sajid.iags@pu.edu.pk
Shahbaz Muhammad 1 2
Shahzad Ahmad Naeem 3
Khan Hafiz Azhar Ali 1
Anees Moazzam 1
Haider Muhammad Saleem 1
Fatima Ammara 4
1 Institute of Agricultural Sciences, University of the Punjab , Lahore , Pakistan
2 Department of Biology, Colorado State University , Fort Collins, Colorado , USA
3 Department of Agronomy, Bahauddin Zakariya University , Multan , Pakistan
4 Department of Environmental Science, Lahore College for Women University , Lahore , Pakistan
Sun Xiaolin
Electronic publication date: 2015 Aug 4
Publication date: 2015
Volume: 3
Electronic Location ID: e1119
Received 2015 Feb 12; Accepted 2015 Jul 2
Copyright: © 2015 Ali et al.
Copyright year: 2015
Copyright holder: Ali et al.
License: This is an open access article distributed under the terms of the Creative Commons Attribution License, which permits unrestricted use, distribution, reproduction and adaptation in any medium and for any purpose provided that it is properly attributed. For attribution, the original author(s), title, publication source (PeerJ) and either DOI or URL of the article must be cited.
License URL: https://creativecommons.org/licenses/by/4.0/

Keywords: Leaf chlorosis, Nutrient uptake, Non-protein thiol, Cu contamination, Biomass, Sulfur, Brassica, Copper, Thiols, Hydroponics, Toxicity, Cabbage

Funding: University of the Punjab, Lahore Funding was provided by the University of the Punjab, Lahore. The funders had no role in study design, data collection and analysis, decision to publish, or preparation of the manuscript.

==============================
Arable soils are frequently subjected to contamination with copper as the consequence of imbalanced fertilization with manure and organic fertilizers and/or extensive use of copper-containing fungicides. In the present study, the exposure of stone-head cabbage (Brassica oleracea var. capitata) to elevated Cu2+ levels resulted in leaf chlorosis and lesser biomass yield at ≥2 µ M. Root nitrate content was not statistically affected by Cu2+ levels, although it was substantially decreased at ≥5 µ M Cu2+ in the shoot. The decrease in nitrate contents can be related to lower nitrate uptake rates because of growth inhibition by Cu-toxicity. Shoot sulfate content increased strongly at ≥2 µ M Cu2+ indicating an increase in demand for sulfur under Cu stress. Furthermore, at ≥2 µM concentration, concentration of water-soluble non-protein thiol increased markedly in the roots and to a smaller level in the shoot. When exposed to elevated concentrations of Cu2+ the improved sulfate and water-soluble non-protein thiols need further studies for the evaluation of their direct relation with the synthesis of metal-chelating compounds (i.e., phytochelatins).

Introduction

Transition metals such as copper (Cu), zinc (Zn) and molybdenum (Mo) are essential for the growth and development of plants, but they rapidly get toxic at higher levels (Kopsell & Kopsell, 2007). Cu contamination in agricultural soils as a consequence of mining metals, dispersal of sewage sludge, arbitrary and improper application of agrochemicals, addition of organic fertilizers and frequent use of irrigation with low quality water is a well-known problem (Dach & Starmans, 2005; Yruela, 2009). Cu is a redox active metal that can exist in both Cu2+ and Cu+ forms in living organisms. At the protein level, Cu serves as a co-factor for various enzymes such as Cu/Zn-superoxide dismutase (Cu/ZnSOD), cytochrome c oxidase, ascorbate oxidase, amino oxidase, laccase, plastocyanin (PC), and polyphenol oxidase (Yruela, 2005; Yruela, 2009; Pilon et al., 2006). However, redox cycling between Cu2+ and Cu+ could induce oxidative stress by producing highly toxic hydroxyl radicals (Yruela, 2005; Yruela, 2009).

Plants exposed to elevated levels of Cu show unspecific toxicity symptoms. Elevated Cu levels in soils primarily result in stunted root growth and leaf chlorosis (Kopsell & Kopsell, 2007; Shahbaz et al., 2010a). Copper toxicity-induced reduction in chlorophyll contents hinder the development of chloroplast, thalakoid membrane and photosystem II (PSII), which are considered as the most sensitive Cu toxicity sites (Pätsikkä, Aroan & Tyystjärvi, 1998; Pätsikkä et al., 2002; Burkhead et al., 2009; Yruela, 2005; Yruela, 2009; Shahbaz et al., 2010b). At cellular level, toxicity may lead to binding of sulfhydryl groups in proteins, insufficiency or excess of other essential ions, oxidative damage and reduced cell transport (De Vos et al., 1993; Yruela, 2009). Furthermore, Cu-toxicity can change the mineral composition of plants. For instance, Fe contents may decrease in the shoot (Pätsikkä, Aroan & Tyystjärvi, 1998; Pätsikkä et al., 2002; Kopsell & Kopsell, 2007; Shahbaz et al., 2010b), Ca and Mg may decrease in the root and Zn contents may increase in both root and shoot upon Cu exposure at elevated levels (Shahbaz et al., 2010b).

Root growth is more severely affected by elevated Cu than shoot growth and the major proportion of Cu uptake retains in the root. Increased Cu contents in the plant tissues induce the synthesis of metal-binding compounds (viz. phytochelatins), which are most likely glutathione-derived compounds. (Inouhe, 2005; Ernst et al., 2008). Inductions of phytochelatins presume that more sulfur is needed for synthesis of these compounds, which results into higher absorption, and incorporation of sulfate. Nonetheless, the role of phytochelatins in detoxification of Cu is not very clear yet (Ernst et al., 2008; Yruela, 2005; Yruela, 2009; Shahbaz et al., 2010a).

Brassica and other vegetable crops are often grown in the surrounding areas of big cities and industrial areas in developing countries like Pakistan, where they may be subjected to air and heavy metals pollution (Yang, Stulen & De Kok, 2006). The direct application of sewage water to vegetables is not only the source of many nutrients, but it is often contaminated with high levels of Cu and other heavy metals. As a result of continues untreated sewage application, heavy metals not only accumulate in the soil but also in vegetables (Younas et al., 1998; Butt et al., 2005). High Cu content in crop plants might not only negatively affect plant growth and functioning, but will also enter the food chain (Brun et al., 2001).

The present study used the hydroponics system which allows very close control over water soluble Cu and other mineral concentrations as compared to soil-grown system. Cabbage is a very important vegetable in all over the world. In Pakistan, cabbage is cultivated on almost 4.9 thousand hectares with 76.7 thousand tonnes annual production (FAO, 2013). The present study was aimed to investigate the response of growth, pigment contents and sulfur metabolism of stone-head cabbage grown in hydroponics to copper exposure.

Material and Methods

Stone-head cabbage (Brassica oleracea var. capitata F1) seeds were germinated to sand in a green house. The seedlings collected at ten days after germination were transferred on an aerated 25% modified Hoagland nutrient solution in a 11 liter container (15.8″L × 10.3″W × 7″H; 3 plants set−1 and 12 sets container−1) in a greenhouse for 10 days. The nutrient solution consists of 1.25 mM Ca(NO3)2.4H2O, 1.25 mM KNO3, 0.25 mM KH2PO4, 0/0.5 mM MgSO4.7H2O, 11.6 µM H3BO3, 2.4 µM MnCl2.4H2O, 0.24 µM ZnSO4.7H2O, 0.08 µM CuSO4.5H2O, 0.13 µM Na2MoO4.2H2O and 22.5 µM Fe3+-EDTA with supplemental concentrations of 0, 2, 5 and 10 µM CuCl2 and pH 5.9–6.0. The nutrient solution was continuously aerated with Aqua-Supreme—Air Pump—Model AP-4.The photoperiod was 14 h. 30 and 25 °C (±5 °C) temperatures were set for day and night respectively, whereas the relative humidity was maintained at 60–70%.

Pigment contents

Whole shoot was homogenized (in 100% acetone 10 mL per g FW) followed by centrifugation at 800 g for 20 min. Lichtenthaler (1987) was followed for the determination of chlorophyll a, b and total carotenoid contents.

Nitrate and sulfate contents

Frozen root and shoot material was homogenized in de-mineralized water (10 mL per g fresh weight) and one layer of Miracloth filter was used to filter the homogenate. The supernatant was incubated in a water bath at 100 °C for 10 min. The remainder was centrifuged for 15 min (0 °C) at 30,000 g. The anions were separated by HPLC and Maas et al. (1986) was followed for their refractometric determination using a Knauer differential refractometer (model 98.00, Bad Homburg, Germany).

Water-soluble non-protein thiols

Extraction medium containing 80 mM sulfosalicylic acid, 1 mM EDTA, and 0.15% (w/v) ascorbic acid with an Ultra Turrax at 0 °C (10 mL per g fresh weight) was used for the homogenization of fresh plant matter. The resultant homogenous material was passed through  one layer of Miracloth which was then centrifuged at 30,000 g for 15 min (0 °C). De Kok, Buwalda & Bosma (1988) was followed for the determination of total water-soluble non-protein thiol content colorimetrically at 413 nm after reaction with 5, 5′-dithiobis [2-nitrobenzoic acid].

Results

Plant biomass in response to Cu exposure

Exposure of stone-head cabbage to higher concentrations of Cu2+ (≥2 µM) in nutrient solution caused chlorosis of both the shoot and young emerging leaves, that ultimately reduced both root and shoot biomass production (Fig. 1). A 10 day exposure to increasing Cu2+ concentrations in nutrient media led to a significant reduction of both root and shoot biomass production at ≥2 µM Cu2+. Shoot to root ratio improved at ≥5 µM Cu2+, demonstrating that when exposed to copper, root growth was more affected than shoot growth (Fig. 1). Root dry matter content increased at 10 µM Cu2+, whereas shoot dry matter content increased at ≥5 µM Cu2+ (Fig. 2).

Figure 1 Impact of elevated levels of Cu2+ on biomass production of stone-head cabbage (Brassica oleracea var. capitata).

10-day-old seedlings of stone-head cabbage (Brassica oleracea var. capitata) were grown on a 25% Hoagland solution containing 0, 2, 5 and 10 µM CuCl2 in the root environment. Data on biomass production (g FW) and shoot/root ratio represent the mean of 2 independent experiments with 9 measurements having 3 plants in each treatment (±SD). Means with different letters differ significantly at p ≤ 0.01 (Student’s t-test).

Figure 2 Impact of elevated levels of Cu2+ on dry matter content of stone-head cabbage (Brassica oleracea var. capitata).

10-day-old seedlings of stone-head cabbage (Brassica oleracea var. capitata) were grown on a 25% Hoagland solution containing 0, 2, 5 and 10 µM CuCl2 in the root environment. Data on dry matter content (%) represent the mean of 2 independent experiments with 9 measurements having 3 plants in each treatment (±SD). Means with different letters differ significantly at p ≤ 0.01 (Student’s t-test).

Pigment content in response to Cu exposure

The total chlorophyll (Chl. a + b) and carotenoid contents of stone-head cabbage were significantly decreased upon exposure at ≥2 µM Cu2+ (Fig. 3). There were significant decreases in chlorophyll a/b and chlorophyll/carotenoid ratios when exposed to increased Cu2+ concentrations (10 µM Cu2+). Ten µM Cu2+ exposure resulted in the start of rapid development of shoot chlorosis and significantly faster reduction in chlorophyll a contents of chlorophyll b and carotenoids, ultimately leading to a significant reduction in chlorophyll a/b and chlorophyll/carotenoid ratios (Fig. 3).

Figure 3 Impact of elevated levels of Cu2+ on pigment content (chl. a + b & carotenoids) of stone-head cabbage (Brassica oleracea var. capitata).

10-day-old seedlings of stone-head cabbage (Brassica oleracea var. capitata) were grown on a 25% Hoagland solution containing 0, 2, 5 and 10 µM CuCl2 in the root environment. Data on chlorophyll content (chl. a + b; mg g−1 FW) and carotenoid content (mg g−1 FW) represent the mean of 2 independent experiments with 9 measurements having 3 plants in each treatment (±SD). Means with different letters differ significantly at p ≤ 0.01 (Student’s t-test).

Sulfate and water-soluble non-protein thiol contents in response to Cu exposure

Elevated Cu2+ levels showed a significant effect on concentration of the nitrate, sulfate and water-soluble non-protein thiol in stone-head cabbage. The nitrate contents of the roots showed a non significant response to the Cu exposure at different levels, however in shoots it were significantly decreased at ≥5 µM Cu2+ (Fig. 4). Sulfate contents in the roots were not affected; however, Cu2+ treatments of ≥2 µM substantially increased the sulfate contents of the shoot (Fig. 4). There was slight decrease in nitrate to sulfate ratio in the root and a strong decrease in the shoot when exposed to elevated levels of Cu (Fig. 4). Furthermore, the exposure to ≥2 µM Cu2+ resulted in a solid raise in water-soluble non-protein thiol contents in the roots and to a smaller degree in the shoots at 10 µM Cu2+ (Fig. 5).

Figure 4 Impact of elevated levels of Cu2+ on pigment content (chl.a/chl.b & chl./car. ratio) of stone-head cabbage (Brassica oleracea var. capitata).

10-day-old seedlings of stone-head cabbage (Brassica oleracea var. capitata) were grown on a 25% Hoagland solution containing 0, 2, 5 and 10 µM CuCl2 in the root environment. Data on chlorophyll content (chl.a/chl.b and chl./carotenoid ratio) represent the mean of 2 independent experiments with 9 measurements having 3 plants in each treatment (±SD). Means with different letters differ significantly at p ≤ 0.01 (Student’s t-test).

Figure 5 Impact of elevated levels of Cu2+ on nitrate and sulfate content of stone-head cabbage (Brassica oleracea var. capitata).

10-day-old seedlings of stone-head cabbage (Brassica oleracea var. capitata) were grown on a 25% Hoagland solution containing 0, 2, 5 and 10 µM CuCl2 in the root environment. Data on nitrate and sulfate content (µmol g−1 FW) represent the mean of 2 independent experiments with 9 measurements having 3 plants in each treatment (±SD). Means with different letters differ significantly at p ≤ 0.01 (Student’s t-test).

Discussion

Cu exposure at elevated levels (<2 µM Cu2+) to stone-head cabbage significantly decreased the production of root and shoot biomass and raised the ratio of the shoot to the root. Copper contamination in the root environment generally results in retarded production of root and shoot biomass and a reduced photosynthetic activity. Moreover, it causes chlorosis, necrosis and bleaching of pigments (Yruela, 2005; Yruela, 2009; Sheldon & Menzies, 2005; Shahbaz et al., 2010a; Shahbaz et al., 2010b). In cabbage, the reduced production of biomass when exposed to elevated Cu levels coincided with decreased pigment contents (chl. a, b, carotenoids; Fig. 3) which may have resulted in reduced activity of photosynthesis and the dark respiration rate (Shahbaz et al., 2010a). It has been shown that Cu-toxicity damages chloroplasts either by inducing iron deficiency or by replacing Mg in the chlorophyll by Cu (Pätsikkä et al., 2002; Küpper et al., 2003). Cu exposure at elevated levels not only decreased the pigment content but there was also a change in pigment composition. Chlorophyll a content decreased significantly faster than that of chlorophyll b and carotenoids, which resulted in a decreased chlorophyll a/b and chlorophyll/carotenoid ratio. Similar results were reported by Chu et al. (2006) in Trifolium repens L.

It is shown that the production of root biomass was more influenced than that of the shoot biomass production. The relatively higher reduction in the root biomass upon exposure to metal contamination could be due the fact that roots come in direct contact with toxic metals (Cd, Cu). Toxic metal-induced hindered root growth also reduces the uptake of essential nutrients (Sheldon & Menzies, 2005).

Plants have evolved a tightly-controlled mechanism for the absorption, allocation and assimilation of sulfate under normal conditions. (Hawkesford & De Kok, 2006). Enhanced exposure of cabbage to Cu concentrations considerably affects the contents and allocation of sulfur compounds in the root and shoot of cabbage (Shahbaz et al., 2010a). The raised shoot sulfate contents might be attributed to Cu-toxicity induced upregulation of the sulfate suppliers in roots (Shahbaz et al., 2010a). To maintain rapid growth rates under stress conditions, Brassica species increase their demand for sulfur supply (Koralewska et al., 2008; Koralewska et al., 2009). Exposure of stone-head cabbage to elevated levels of Cu did not affect the nitrate contents in roots, however at ≥5 µM Cu2+ there was a significant decrease in the shoot nitrate contents. The reduced production of plant biomass at ≥5 µM Cu2+ could be attributed to reduced supply of nitrate in the shoot. The decrease in nitrate: sulfate ratio in the shoot of stone-head cabbage under elevated Cu levels may be attributed to enhanced sulfate contents, however the link between uptake rates of nitrate and sulfate is not evident yet (Stulen & De Kok, 2012). Since both nitrate and sulfate are involved in amino acid and protein synthesis, their uptake rates are related with growth rates (Stulen & De Kok, 2012).

Roots accumulated a slightly raised level of water-soluble non-protein thiols when compared with that of the shoot. Only a small proportion of the rise in thiol content might be attributed to a Cu-induced synthesis of phytochelatins in Chinese cabbage (Shahbaz et al., 2010a). Previous reports suggest that the formation of sulfur-rich metal-chelating compounds (i.e., water-soluble non-protein thiols) when disclosed to prospective toxic metals may perhaps require an enhanced demand for sulfur, viz. the absorption rate and incorporation of sulfate (Sirko & Gotor, 2007; Ernst et al., 2008). The possible significance of sulfur nutrition upon elevated copper exposure needs further investigation.

Conclusions

This investigation has shown that the elevated copper levels in the root surroundings proved toxic for stone-head cabbage. Copper exposure at ≥2 µM Cu2+ negatively affected the plant biomass production and pigment contents. Furthermore, elevated Cu content considerably affected the concentration of sulfate and water-soluble non-protein thiol of stone-head cabbage which might be due to the induction of phytochelatins to detoxify excess copper.

Supplemental Information

Supplemental Information 1 Means and Standard Devations of the Prodeced Data

Click here for additional data file.

Additional Information and Declarations

Competing Interests

Author Contributions

The authors declare there are no competing interests.

Sajid Ali conceived and designed the experiments, wrote the paper, prepared figures and/or tables, reviewed drafts of the paper.

Muhammad Shahbaz conceived and designed the experiments, performed the experiments, wrote the paper.

Ahmad Naeem Shahzad and Ammara Fatima analyzed the data.

Hafiz Azhar Ali Khan contributed reagents/materials/analysis tools, wrote the paper, prepared figures and/or tables, reviewed drafts of the paper.

Moazzam Anees contributed reagents/materials/analysis tools.

Muhammad Saleem Haider provided the facilities (laboratory and staff) for the research work.

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
