# Peer review of "Impact of copper toxicity on stone-head cabbage (Brassica oleracea var. capitata) in hydroponics"

_PeerJ, doi:10.7717/peerj.1119_

## Round 0.1 · original submission · Major Revisions

This manuscript describes the copper toxicity effect on Brassica oleracea var. capitata. The methods used and basic results are technically sound. However, there is a large room for further improvement to make it publishable. Please carefully check out and satisfy each of questions and confusions found by the two reviewers in current manuscript, especially for all format-related issues and re-writing the Discussion section based on feedback of the reviewers. Please provide a letter for point-by-point answers and corrections to each of the questions and comments when you submit your revised version.

Reviewer 1 ·

Basic reporting

1. The manuscript was not written in accordance with the guidelines and requirements set by PeerJ, in terms of the templates and formats (text and references). Some problematic areas are listed below:
No sequential line numbering is provided;
No keywords are listed;
Wrong symbols are used in the abstract, such as Cu2+;
Reference cited in the text cannot be found in References section, for example, Caspi et al., 1999;
Incorrect in-text citation and confusing formatting of references listed in References section;
2. Suggest authors take time to carefully and thoroughly read submission guidelines from PeerJ website and to revise the manuscript by following all recommendations and instructions given by PeerJ.
3. Authors should add some information regarding copper toxicity on vegetable crops in Introduction section.
4. All figures are not presented and labelled appropriately. What is the unit in Y axis? How can you share shoot/root ratio and shoot biomass (g FW) in same unit in Y axis (Figure 1)? How did you perform the Student’s t-test for multiple comparisons of the treatments? For several treatment groups, a better approach is to use ANOVA and get a pooled estimate of error provided the errors appear to be homogeneous across treatment groups. This way you will get a more reliable error estimate and a comparison of means.

Experimental design

1. Materials and methods are not described with sufficient information, for example, what is the source of the chemicals used in the experiments; what size of the containers are used; in which season the experiments were conducted; what type and model of instrument or device is used for the measurement and test; what type of hydroponics is used; etc.
2. CuSO4 is a common source for conducting plant nutrient experiments in hydroponics. Why do you use CuCl2 instead of CuSO4? Please keep in mind that high Cl- in growth solution may cause some unnecessary complications for this kind of experiment.

Validity of the findings

1.The data should be analysed as suggested in the basic reporting.
2. I did not see any explanation and discussion about the data of chlorophyll (a + b) and chlorophyll/carotenoid shown in Figure 4. If authors think the data of Figure 4 should be included in the manuscript then you need to give the explanation and discussion of the data shown the figure.
3. It would be good if authors could provide one image to show the chlorosis leaves and the impact of shoot and root growth in different concentrations of copper.

Reviewer 2 ·

Basic reporting

The article is written in English following professional standards. It includes background, methods, results and discussion sections. Figures are clear.

Experimental design

The investigation is well conducted and methods describe sufficient information.

Validity of the findings

This article provide data on Copper toxicity effect on Brassica oleracea var. capitata. The results are clear and similar to those found in other species. Authors should be discuss the results in the context of other Brassica species or other cultivars. The article is concise. Considering that copper toxicity has been extensively investigated the authors should be compare and conect the results with previous data reported in order to give information of posible differences or particularities in this cultivar.

Additional comments

I consider that this article should be revised, in particular the discussion section.

---

## Round 0.2 · Minor Revisions

This revised version still contains some typos (for example "Piment", I think you want to say pigment). It is not difficult to check the spelling given that the word document has automatic spell check. Again, as pointed out by the reviewer 1 - there are lot of missing information and errors in Reference, Table and Figures which should have not been happened in the revised manuscript. The authors must VERY CAREFULLY check and make corresponding corrections to each of the points raised by me and the reviewers.

Reviewer 1 ·

Basic reporting

1. The references cited in the text do not appear in the References section.
Page 4, line 73, where is the “Table2”?
Page 7, line 167, there are no “Koralewska et al., 2008; 2009” in the References section.
2. Figure number is mistakenly cited in the text.
Page 6, line 129, Fig. 3?
Page 6, line 134 and 137, Fig. 4?
3. All figures are not presented and labelled appropriately. What is the unit in Y axis? How can you share shoot/root ratio and shoot biomass (g FW) in same unit in Y axis (Figure 1)?
The revised manuscript did not make any changes to the issues of the figures. I attached a sample of the right figure presentation from a published article in PeerJ for your reference. Please re-draw Fig. 1 and 4 as the way in the sample given below. Also, there are some missing words in the legends for all five figures (page 12, 14, 16, 18 and 20).

Experimental design

No comments

Validity of the findings

no comments

Annotated reviews are not available for download in order to protect the identity of reviewers who chose to remain anonymous.

---

## Round 0.3 · accepted · Accept

This revised manuscript will now be processed for publication.